# Comparison of Tongue Characteristics Classified According to Ultrasonographic Features Using a K-Means Clustering Algorithm

**DOI:** 10.3390/diagnostics12020264

**Published:** 2022-01-21

**Authors:** Ariya Chantaramanee, Kazuharu Nakagawa, Kanako Yoshimi, Ayako Nakane, Kohei Yamaguchi, Haruka Tohara

**Affiliations:** 1Department of Preventive Dentistry, Naresuan University, Phitsanulok 65000, Thailand; tam_jantra@hotmail.com; 2Department of Dysphagia Rehabilitation, Graduate School of Medical and Dental Sciences, Tokyo Medical and Dental University (TMDU), Tokyo 113-8510, Japan; k.yoshimi.gerd@tmd.ac.jp (K.Y.); a.nakane.swal@tmd.ac.jp (A.N.); yanma627@yahoo.co.jp (K.Y.); harukatohara@hotmail.com (H.T.)

**Keywords:** artificial intelligence, ultrasonography, tongue, algorithm, dysphagia

## Abstract

The precise correlations among tongue function and characteristics remain unknown, and no previous studies have attempted machine learning-based classification of tongue ultrasonography findings. This cross-sectional observational study aimed to investigate relationships among tongue characteristics and function by classifying ultrasound images of the tongue using a K-means clustering algorithm. During 2017–2018, 236 healthy older participants (mean age 70.8 ± 5.4 years) were enrolled. The optimal number of clusters determined by the elbow method was 3. After analysis of tongue thickness and echo intensity plots, tongues were classified into three groups. One-way ANOVA was used to compare tongue function, tongue pressure, and oral diadochokinesis for /ta/ and /ka/ in each group. There were significant differences in all tongue functions among the three groups. The worst function was observed in patients with the lowest values for tongue thickness and echo intensity (tongue pressure [P = 0.023], /ta/ [P = 0.007], and /ka/ [P = 0.038]). Our results indicate that ultrasonographic classification of tongue characteristics using K-means clustering may aid clinicians in selecting the appropriate treatment strategy. Indeed, ultrasonography is advantageous in that it provides real-time imaging that is non-invasive, which can improve patient follow-up both in the clinic and at home.

## 1. Introduction

Several recent studies have investigated the use of ultrasonography for evaluating the muscles of the head and neck, as it enables assessment of both muscle quality and quantity [1]. The tongue is the major organ involved in normal oropharyngeal swallowing [2], consisting of four intrinsic muscles (superior longitudinal, inferior longitudinal, transversus, and verticalis) and four extrinsic muscles (palatoglossus, genioglossus, hyoglossus, and styloglossus) [3], which serve to move and alter the shape of the tongue, respectively [4]. Given that the tongue consists of eight unique muscles, ultrasonography represents an effective strategy for investigating its characteristics in detail.

In addition to qualitative characteristics such as tongue thickness (TT) and cross-sectional area, ultrasonography can be used to assess qualitative characteristics of the tongue, such as the presence of intramuscular adipose tissue and muscle density. These qualitative parameters are represented in terms of echo intensity (EI) on grayscale ultrasonography images [5,6]. One recent study reported that lower EI values are associated with decreased tongue function and increased TT [7], while another identified decreased tongue EI as an independent risk factor for sarcopenic dysphagia in older adults [8]. However, despite a few relevant studies, the precise correlations among EI, TT, and tongue function remain unknown. We hypothesized that ultrasonography images of the tongue would provide insight into these relationships.

However, tongue classification based on ultrasonography is challenging due to the complicated structure of the tongue [9]. As such, researchers have investigated various strategies to aid in classification, including the use of linear classifiers (logistic regression), decision trees (random-forest analysis), support vector machines, and clustering algorithms [10]. Among these, clustering is widely utilized given its simplicity and efficiency [11]. Recent studies have reported that clustering algorithms are highly accurate in distinguishing malignant and benign brain tumors with 95% confidence [12]. For example, Ding et al. reported an accuracy of 91.07% and area under the curve of 0.96 when a clustering algorithm was used to classify breast tumors on ultrasonography [13] (*p* < 0.05). Übeyli and Doğdu also reported that a clustering algorithm could be used to classify erythemato-squamous disease into five categories with an accuracy of 94.22% [14]. 

In the present study, we aimed to investigate the relationships between tongue characteristics and tongue function, including tongue pressure (TP) and diadochokinesis (OD), by classifying ultrasound images of the tongue. Among the various clustering methods available (e.g., K-means clustering, hierarchical clustering, Gaussian mixture models, and density-based clustering [10,15]), we selected K-means clustering because we used two parameters (EI and TT) to describe the characteristics of the tongue, making this method simple and efficient [11]. 

## 2. Materials and Methods

### 2.1. Sample Size

The sample size was calculated using G*Power 3.1 (Kiel University, Kiel, Germany). The alpha value (α, probability of a type I error) and power (1-β, probability of not making a type II error) were set to 0.05 and 0.90, respectively. For this study, we selected a medium effect size of 0.25 [16,17]. The calculation indicated a required sample size of 207 participants across the three groups. 

### 2.2. Participation

The participants included 236 healthy older individuals (71 men, 165 women; mean age: 70.8 ± 5.4 years) from Oyama City (Tochigi, Japan) recruited during 2017–2018. All participants self-reported normal swallowing function and understanding following an explanation of the study. Individuals with a history of neurologic disease, cognitive dysfunction, head and neck cancer or surgery, or any problems related to swallowing function were excluded. Demographic data such as age, sex, weight, height, and body mass index were recorded for each patient.

All study participants provided written informed consent. The study protocols conformed to the guidelines outlined in the Declaration of Helsinki and were approved by the ethics committee of the Faculty of Dentistry at Tokyo Medical and Dental University (D2014-047).

### 2.3. Assessment of Tongue Characteristics

Tongue ultrasonography was performed in Brightness mode using a portable ultrasound machine (M-Turbo; Fujifilm SonoSite, Tokyo, Japan) equipped with a convex transducer (5–10 MHz). All ultrasound examinations were performed by one well-trained examiner, with the participant in a relaxed, seated position. For measurement, the probe was placed underneath the chin, and the angle of the probe was positioned perpendicular to the Frankfurt horizontal plane at the first premolar area (Figure 1A,B) [7,8,9] using passive pressure. Echo gain was maintained at the same level for all measurements, which were obtained with the tongue in the resting position after swallowing saliva. This process was repeated thrice, and the mean of the three values was recorded for each measurement.

TT and EI measurements were analyzed using ImageJ (version 1.37, National Institutes of Health, Rockville, MD, USA). TT was measured from the dorsal surface of the tongue to the upper border of the geniohyoid muscle. A region of interest that included as much tongue muscle tissue as possible while avoiding the surrounding fascia was selected to determine EI (Figure 1C) [7]. The mean EI was measured via a histogram-based grayscale analysis, with values ranging from 0 (black) to 255 (white).

### 2.4. Assessment of Tongue Function

Tongue function was described in terms of tongue strength and tongue skill. Tongue strength was measured as the maximum TP using a JMS TP manometer (JMS Co. Ltd., Tokyo, Japan). An air-filled balloon probe was placed on the dorsal aspect of the tongue. The participant was then instructed to raise the tongue and compress the balloon toward the palate as forcefully as possible [7,18]. Three measurements were obtained, and the average value was recorded as the maximum TP.

Tongue skill was measured based on OD using an oral cavity function testing device. The device, which had a built-in microphone, was placed in front of the mouth, following which the participant was instructed to repeat each of two syllables (/ta/ or /ka/) as quickly as possible for 5 s. This requires using the middle portion and the base of the tongue, respectively [7,19]. The device automatically counted the total number of appropriately pronounced syllables. Repetition speed was calculated as repetitions per second.

### 2.5. Statistical Analysis

Data were analyzed using RStudio version 1.1423 (Rstudio Inc., Boston, MA, USA) and are presented as the mean ± standard deviation. The Kolmogorov–Smirnov test was used to verify that the data followed a normal distribution. Mean values for tongue function (including TP, /ta/, and /ka/) were compared between groups using one-way analysis of variance. Multiple comparisons were performed using Tukey’s test. The level of significance was defined as *p* < 0.05.

Intraclass correlation coefficients (ICC) were used to assess the reliability of the TT and EI measurements. The ICCs were 0.755 and 0.765 for TT and EI, respectively. All intraclass correlation coefficient values were >0.75, indicating good reliability, with values ≥ 0.9 indicating excellent reliability.

### 2.6. Classification Using K-Means Clustering Algorithms

K-means clustering is performed using machine learning algorithms, which learn from input data and use statistical analyses to predict outcomes or perform specific tasks, without requiring explicit instructions [10].

The K-means clustering algorithms are unsupervised clustering algorithms that classify input data-points into classes based on their inherent distance from each other (i.e., centroid-based clustering). When the number of clusters is fixed to K clusters or groups (Figure 2A), initial k centroids (center of the group) are randomly created and placed onto the data plot (Figure 2B), following which the Euclidean distance from each data-point to the centroids is calculated (Equation (1)) (Figure 2C,D) [10,20].
(1)$$\;\;d_{euc}\;\left(x,y\right)=\sqrt{\overset{n}{\underset{i=1}{\sum }}{\left(x_{i}-y_{i}\right)}^{2}}\;\;\;$$
where *x* and *y* are the two vectors of length *n*.

In this study, each data-point was classified into a group according to its closest centroid, based on the Euclidean distance between the data-point and the centroid (Figure 2E). After the first classification, the centroid was updated to the new location (i.e., the actual center of the group) based on the mean value of all data-points in the group (Figure 2F), following which the distance between each data-point and the new centroid was calculated (Figure 2G). These steps were repeated until the mean value of all data-points stopped changing (i.e., the new centroids remained in the same location) (Figure 2H) [10,15]. 

The most popular methods for determining the optimal number of clusters are the silhouette method and the elbow method—the latter of which was used in this study [21]. This method aims to minimize the sum of the square of the Euclidean distances between each point and its corresponding centroid (total intra-cluster variation, also known as total within-cluster sum of squares (*tot.withinss*; Equation (2)). A smaller value for *tot.withinss* indicates that the data-points are close to the centroid; therefore, it measures the compactness of the clustering and should ideally be as small as possible [10,20].
(2)$$\;\;tot.withinss=\overset{k}{\underset{i=1}{\sum }}\underset{x_{i}\in C_{k}}{\sum }{\left(x_{i}-\mu_{i}\right)}^{2}\;\;$$
where *k* is the number of clusters, *C* is the cluster (*C* = *C*2, *C*3, …, *Ck*), *x_i_* is a data-point belonging to cluster *C_k_*, and *µ* is the mean value of the data-points assigned to cluster *C_k_*.

For the optimal cluster calculation, the data were first assessed using K-means clustering algorithms in which *k* varied from 1 to 10. Then, *tot.withinss* was calculated for each *k* and plotted according to the number of clusters, *k* (Figure 3). The location of a bend in the plot is generally considered to indicate the appropriate number of clusters because it indicates that adding another cluster does not markedly improve *tot.withinss* [15,22,23]. Figure 3 shows that the bend occurred at three clusters (*k* = 3).

In this study, we plotted EI on the *x* axis and TT on the *y* axis (Figure 4). The “kmeans” (d, centers) function in R software was used to calculate the Euclidean distance, locate the centroid, and repeat all steps of the calculation. In this context, “d” refers to the numeric matrix of data (EI, TT), while “centers” refers to the number of clusters (i.e., 3).

## 3. Results

### 3.1. Determining Optimal Clusters

The plot exhibited a sharp bend at three clusters (*k* = 3), indicating the optimal number of clusters for the dataset (Figure 3). Therefore, tongue characteristics were classified into three groups (Group 1, Group 2, and Group 3).

### 3.2. Tongue Characteristics

EI and TT data were plotted and divided into three groups using K-means clustering algorithms (Figure 4). Average TT and EI were 37.6 ± 3.7 mm and 55.1 ± 4.7 in Group 1, 40.5 ± 3.5 mm and 42.8 ± 3.5 in Group 2, and 44.0 ± 4.2 mm and 32.2 ± 4.1 in Group 3, respectively (Table 1). These findings indicate that Group 1 exhibited decreased TT and increased brightness when compared with Groups 2 and 3. The tongues of participants in Group 3 were the thickest and darkest (Figure 5).

### 3.3. Tongue Function

TP was highest in Group 2 (32.3 ± 7.1 kPa) and lowest in Group 1 (28.7 kPa), with a significant difference between the two groups (*p* < 0.023). However, there was no significant difference in TP between Groups 2 and 3 (Figure 4).

The average /ta/ and /ka/ values (6.1 and 5.8 time/sec, respectively) were highest in Group 2, and the differences between Groups 2 and 1 were significant (*p* < 0.007 and *p* < 0.038, respectively). The average /ta/ value was also greater in Group 3 than in Group 1 (*p* < 0.012), which had the lowest average /ta/ and /ka/ values (5.6 and 5.4 times/sec, respectively). However, the average /ta/ and /ka/ values did not significantly differ between Groups 2 and 3 (Figure 4).

Overall, the findings indicate that tongue function was poorest in Group 1, but that there was no significant difference in tongue function between Groups 2 and 3.

## 4. Discussion

In this study, we utilized K-means clustering to classify patterns of tongue characteristics based on ultrasound measurements. Our findings indicated that participants in Group 1 exhibited the poorest tongue function in terms of both TP and OD. Moreover, our analysis suggests that K-means clustering is useful for predicting tongue function based on ultrasonography findings.

### 4.1. Relationship between Tongue Group and TP

Group 1 exhibited the lowest values for TT and TP and the highest value for EI. TP is an important indicator of tongue muscle strength and swallowing during the oral phase [24,25]. Previous studies have demonstrated a correlation between TT and TP, which is plausible given that muscle mass is commonly associated with muscle strength [26,27]. EI reflects intramuscular adipose tissue content: Higher EI indicates greater adiposity, which may affect tongue function, as indicated by the findings in Group 1 [7,28,29,30]. Thus, when the tongue appears thinner and brighter on ultrasonography, as noted in Group 1, tongue strength is likely to be lower.

However, another study suggested that EI itself may not be related to TP [7]. This discrepancy may be explained by differences in study design. The authors of the previous study used regression analysis to identify factors that could predict EI, which identified TT and OD as the only significant factors. However, the present study classified ultrasonographic tongue characteristics into three groups, following which TP was compared among the groups.

### 4.2. Relationship between Tongue Groups and OD

OD refers to the rate of articulation, reflecting the speed of tongue movement [31]. Group 1 exhibited the lowest TP and OD values. Both TP and OD are commonly used to determine the efficacy of speech production. Previous research has highlighted the relationship between strength and speed during speech production. Hence, TP and OD should be related, as observed in the present study [2,31,32]. Although muscle mass is clearly associated with muscle strength, no previous studies have reported a direct relationship between TT and OD. However, one study noted a relationship between EI and /ta/ or /ka/ [7].

Based on the above, patients in Group 1 should exhibit the poorest tongue function, followed by those in Group 2, while patients in Group 3 should exhibit the best tongue function. However, in our study, patients in Group 2 exhibited the strongest tongue function, except for /ta/ (Table 1). Furthermore, there were no statistically significant differences between Groups 2 and 3. This finding suggests that muscle quantity and quality may be sufficient for maintaining tongue function in Groups 2 and 3, but not in Group 1. Further studies are required to fully elucidate the relationships between tongue characteristics and tongue function.

### 4.3. Categorizing Tongue Characteristics Using K-Means Clustering Algorithms

Previous studies have investigated various methods for applying medical image segmentation [12,13,14,33]. Generally, segmentation methods are used to detect diseases such as brain cancer based on magnetic resonance images [12,33], breast cancer based on ultrasonographic images [13], or skin diseases based on clinical features [14]. While these studies have highlighted the accuracy of K-means clustering, the final diagnosis of all medical images was known, and the clustering method was mainly used to detect disease based on imaging, following which the findings were compared with the true diagnosis to establish the accuracy of the method. However, as we did not know the diagnosis in each case, we were unable to determine the accuracy of the current classification method.

Several studies have investigated whether EI can be used for tongue classification. The EI of the tongue increases with age [34]. Research has also indicated that the EI of the tongue is significantly higher in patients with amyotrophic lateral sclerosis (ALS) than in healthy participants [35], suggesting that EI can be used to distinguish between healthy and diseased tongue muscle. However, no previous studies have examined these associations using detailed classifications based on ultrasonography or significant differences in tongue function to define each group. Our findings indicate that K-means clustering may be an effective approach for classification of the tongue. Future studies should aim to collect data for various patient groups based on age, sex, and disease status, as this may help to determine whether our method can be used to improve diagnosis.

### 4.4. Clinical Implications

Several studies have indicated that older individuals are more likely to experience dysphagia due to age-related decreases in muscle mass [26,36,37,38]. Thus, assessments of muscle mass and the strength/function of the perioral muscles are necessary for maintaining oral function in older adults. Our analysis indicated that patients in Group 1 exhibited the thinnest, brightest, and most easily detectable tongues on ultrasonography (Figure 5). Establishing standard diagnostic criteria based on ultrasonography patterns may represent a more useful and non-invasive strategy for assessing and preventing oral hypofunction. Furthermore, identifying the risks associated with each ultrasonography pattern may aid in determining the appropriate treatments and exercises for community-dwelling older adults. Indeed, one recent study demonstrated that training the tongue by pushing it against a hard palate, (i.e., tongue-pressure resistance training) can improve both tongue (TP and OD) and suprahyoid muscle function [37]. Another study suggested that 3 months of oral training can improve swallowing function and OD in older adults who are at high risk of deterioration in oral health [39].

Furthermore, aging leads to atrophy of the tongue papillae, and several studies have reported an association between tongue function and the characteristics of the tongue surface [40,41]. Appropriate diagnostic assessment of tongue function can aid clinicians in selecting the proper treatment strategy, which may in turn aid older adults in maintaining oral hygiene, feeding ability, and swallowing function. Given that these strategies can help to prevent sarcopenic dysphagia, our classification method has important implications regarding quality of life in older individuals.

### 4.5. Limitations

Our study had some limitations. First, the dataset used for training the classification model was based on only 236 participants, which may have been insufficient for determining the outcome without error. Thus, a larger sample size required to verify our model. Second, all study participants were healthy; none of them had compromised tongue function or neurological status. As such, we were unable to investigate the effects of aging and disease such as neuromuscular diseases on tongue characteristics. In addition, muscle fibrosis and adiposity are major characteristics of sarcopenia, and in general, age-related fibrosis of muscle tissue is also associated with increased EI [42]. In our method, we categorized the participants’ tongue using EI and TT. Therefore, the influence of aging on EI could not be eliminated. Furthermore, it is difficult to distinguish diseases such as neuromuscular disorders using our classification. Longitudinal studies that include participants with poor tongue function, systemic sarcopenia, and various diseases may aid in determining the usefulness of our classification method.

## 5. Conclusions

We used K-mean clustering algorithms on ultrasonographic images to categorize tongue characteristics based on muscle luminance and TT of healthy older individuals. In this study, the elbow method was used to calculate *tot.withinss* for each *k* and plotted it according to the number of clusters. The optimal number of clusters determined by this method was three. EI was plotted on the x-axis and TT on the y-axis for analysis, and the subjects were classified into three groups.

The results showed that Group 1, which had the highest EI, had the lowest TT and TP and significantly lower OD. The classification of tongue images using K-mean clustering algorithms could be applied for predicting tongue function and for diagnosis. If used by clinicians as a tool to prevent the decline of oral and swallowing functions, it may help to provide functional training and follow-up for older adults. For accurate classification, more sample analysis is needed in the future to establish an algorithm.

Ultrasonography is not only easy to use for real-time imaging but is also radiation-free, painless, portable, and can be used not only in hospitals but also in home-visits. However, it is difficult at present to distinguish the cause of changes in EI from the images. EI is affected by aging and diseases, as well as sarcopenia. The limitation of this study is that the effect of aging cannot be excluded. Therefore, it is necessary to analyze the imaging characteristics of the tongue and related factors for various ages and diseases in the future.

## Figures and Tables

**Figure 1 diagnostics-12-00264-f001:**
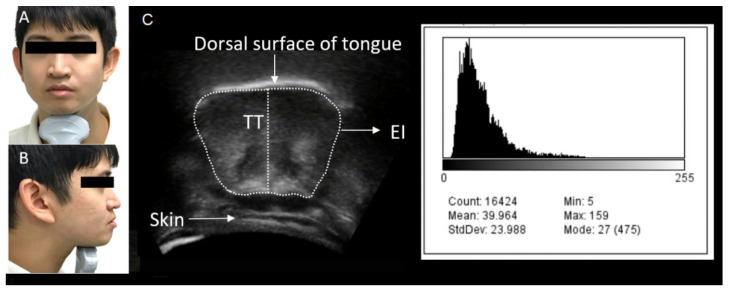
Points of measurement. (**A**) Position of the ultrasonography probe (anterior view). (**B**) Position of the ultrasonography probe (lateral view). (**C**) Ultrasonographic image and grayscale histogram of the tongue.

**Figure 2 diagnostics-12-00264-f002:**
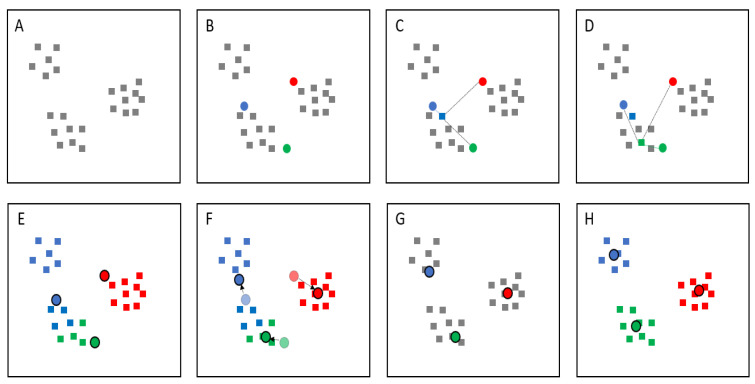
Demonstration of the K-means clustering algorithm, *k* = 3. (**A**) Data plot. (**B**) Three centroids are randomly created, and (**C**) the distance from each data-point to the centroids is calculated using Euclidean distance. (**D**) The step is repeated for all data-points, (**E**) each of which is classified into a group according to its closest centroid. (**F**) The centroid is updated to the new location based on the mean value of all data-points in the group. (**G**) The distance between each data-point and the new centroids is calculated again. (**H**) These steps are repeated until the mean value of all data-points stops changing.

**Figure 3 diagnostics-12-00264-f003:**
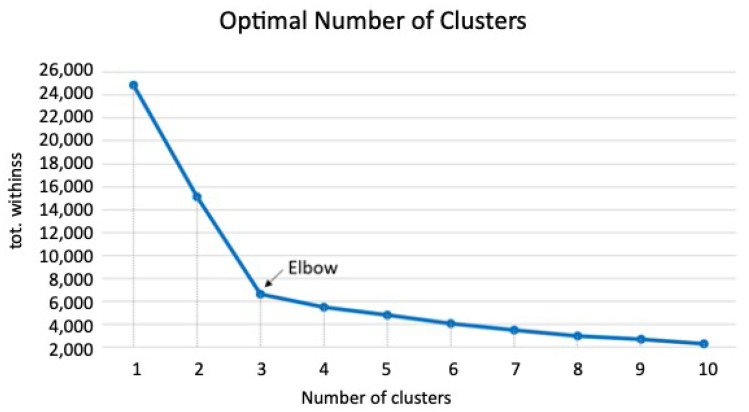
Linear plot between *tot.withinss* and the number of clusters (varying from 1 to 10). The bending point is located at *k* = 3, which represents the optimal number of clusters.

**Figure 4 diagnostics-12-00264-f004:**
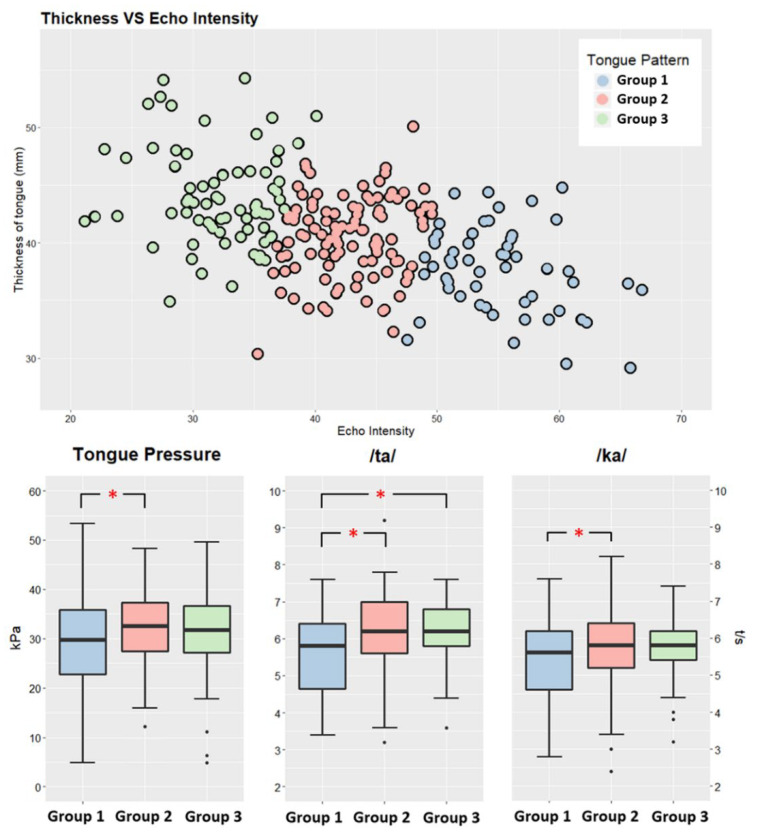
Scatter plot of echo intensity and tongue thickness using the K-means for the three groups. Box plot comparison of tongue function (tongue pressure, /ta/, and /ka/) among the groups based on tongue characteristics. Tongue pressure: Group 1 vs. Group 2, P = 0.023 (* *p* value < 0.05). /ta/: Group 1 vs. Group 3, P = 0.001 (* *p* value < 0.05); Group 1 vs. Group 2, P = 0.007 (* *p* value < 0.05). /ka/: Group 1 vs. Group 2, P = 0.038 (* *p* value < 0.05).

**Figure 5 diagnostics-12-00264-f005:**
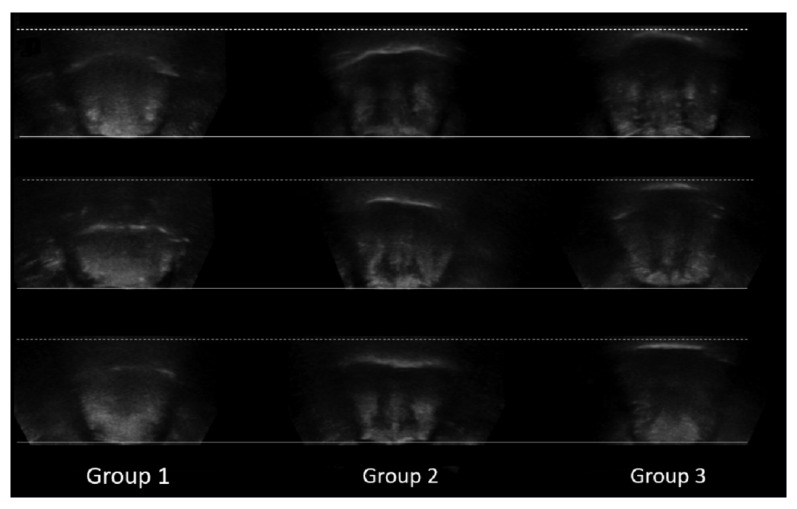
Comparison of illustrative ultrasound images in each group. The solid and dashed lines represent the lowest and the highest parts of the tongue (dorsal surface of the tongue) in Group 3, respectively.

**Table 1 diagnostics-12-00264-t001:** Participant characteristics (*n* = 236).

Variables	Group 1 (Mean ± Standard Deviation)	Group 2 (Mean ± Standard Deviation)	Group 3 (Mean ± Standard Deviation)	Range	*p*-Value (ANOVA ^†^)
Physical data					
Number	54	109	73	-	-
Sex (female, %)	71.6	74.1	64.4	-	-
Age (years)	72.6 ± 5.0	69.8 ± 5.6	71.0 ± 5.2	65.0–86.0	0.007
BMI ^‡^ (kg/m^2^)	23.4 ± 2.9	22.7 ± 2.9	22.4 ± 2.7	14.0–32.4	0.154
Ultrasonographic data					
Tongue thickness (mm)	37.6 ± 3.7	40.5 ± 3.5	44.0 ± 4.2	29.2–54.3	<0.001
Echo intensity	55.1 ± 4.7	42.8 ± 3.5	32.2 ± 4.1	21.1–66.8	<0.001
Tongue function data					
Tongue pressure (kPa)	28.7 ± 9.9	32.3 ± 7.1	31.4 ± 8.2	4.9–53.3	0.030
/ta/	5.6 ± 1.1	6.1 ± 1.2	6.2 ± 0.8	3.2–9.2	0.005
/ka/	5.4 ± 1.1	5.8 ± 1.1	5.7 ± 0.8	2.4–10.2	0.040

^†^ ANOVA, analysis of variance; ^‡^ BMI, body mass index.

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
