# Peer review of "Comparison of Tongue Characteristics Classified According to Ultrasonographic Features Using a K-Means Clustering Algorithm"

_diagnostics, 2022, doi:10.3390/diagnostics12020264_

Round 1

Reviewer 1 Report

The paper deals with an interesting topic and is expected to be developed in the future.
We request that the following points be corrected.

1.

I don't think you can exclude the effect of aging because you didn't do a final multivariate analysis to draw this conclusion. Please add a multivariate analysis.

2.

In general, aging causes fibrosis of muscles and fatness of muscle tissue. The point of age-related changes is not mentioned in the discussion.

Even in normal aging, there is an increase in Echo intensity due to fibrosis (A). It has also been noted that in pathological muscle tissue, increased Echo intensity is indicative of muscle fiber entrapment (B).

Consideration should be given to the fibrosis and adiposity of the muscle, which can be confirmed by ultrasonography.

A; Sigrid Pillen, Ilse M P Arts, Machiel J Zwarts. Muscle ultrasound in neuromuscular disorders . Muscle Nerve. 2008 Jun;37(6):679-93. doi: 10.1002/mus.21015.

B; Nicoletta Battisti, David Milletti, Marco Miceli, et al. Usefulness of a Qualitative Ultrasound Evaluation of the Gastrocnemius-Soleus Complex with the Heckmatt Scale for Clinical Practice in Cerebral Palsy. Ultrasound Med Biol. 2018 Dec;44(12):2548-2555. doi: 10.1016/j.ultrasmedbio.2018.08.006.

3.

The diagnostic difficulty with ultrasonography is that the evaluation of Echo intensity is subjective to the evaluator. Please discuss the difficulties in distinguishing between Echo intensity changes due to muscle disease and disuse conditions where active training is recommended.

4.

There is no "4. Discussion". Please structure it properly.

Is "6. Conclusion" after "4.5 Limitations" a mistake for "5. Conclusion"?

Author Response

Comments for Reviewer 1

The paper deals with an interesting topic and is expected to be developed in the future.
We request that the following points be corrected.

Comment 1.

I don't think you can exclude the effect of aging because you didn't do a final multivariate analysis to draw this conclusion. Please add a multivariate analysis.

Response to comments:

Thank you for your comment. As you pointed out, we did not perform a multivariate analysis. There are several reasons for this, as follows.

If the dependent variable is an “ordinal scale”, we use ordinal logistic regression analysis, or if the dependent variable is a “nominal scale with three or more values”, multinominal logistic regression analysis is selected. In this study, Group 1, 2 and 3 should be entered as dependent variables. However, these (Group 1, 2 and 3) are not “ordinal scales”. Furthermore, since the most adequate number of clusters (a number of Group) was calculated using K-means clustering algorithms and linear plot between tot.withinss and the number of clusters, it is not considered to be a “nominal scale” either. Thus, we could not use logistic regression analysis and used ANOVA.

However, as you pointed out, this analysis method cannot eliminate the effect of aging. For further discussion and revision on the effect of aging on EI of the tongue, we have described in comment 2 and 3.

Comment 2.

In general, aging causes fibrosis of muscles and fatness of muscle tissue. The point of age-related changes is not mentioned in the discussion.

Even in normal aging, there is an increase in Echo intensity due to fibrosis (A). It has also been noted that in pathological muscle tissue, increased Echo intensity is indicative of muscle fiber entrapment (B).

Consideration should be given to the fibrosis and adiposity of the muscle, which can be confirmed by ultrasonography.

A; Sigrid Pillen, Ilse M P Arts, Machiel J Zwarts. Muscle ultrasound in neuromuscular disorders . Muscle Nerve. 2008 Jun;37(6):679-93. doi: 10.1002/mus.21015.

B; Nicoletta Battisti, David Milletti, Marco Miceli, et al. Usefulness of a Qualitative Ultrasound Evaluation of the Gastrocnemius-Soleus Complex with the Heckmatt Scale for Clinical Practice in Cerebral Palsy. Ultrasound Med Biol. 2018 Dec;44(12):2548-2555. doi: 10.1016/j.ultrasmedbio.2018.08.006.

Response to comment:

Thank you for pointing this out. We did not discussed about the effect of aging on Echo intensity. In our previous study, we examined EI of the tongue in healthy older adults, there was no significant relationship between EI and age in both the middle and base of tongue (reference No. 7). However, another study suggested an association between tongue EI was related to age (reference No. 34). Although several studies have investigated EI of skeletal muscle, there are few studies focusing on oral region such as tongue muscle. We consider that it is necessary to analyze the effects of aging in the future. We have added these information in the “Discussion” section including “4.5 Limitations” citing the reference you suggested (reference No. 42), along with our response to comment 3.

”Several studies have investigated whether EI can be used for tongue classification. The EI of the tongue increases with age [34]. Research has also indicated that the EI of the tongue is significantly higher in patients with amyotrophic lateral sclerosis (ALS) than in healthy participants [35], suggesting that EI can be used to distinguish between healthy and diseased tongue muscle. However, no previous studies have examined these associations using detailed classifications based on ultrasonography or significant differences in tongue function to define each group. Our findings indicate that K-means clustering may be an effective approach for classification of the tongue. Future studies should aim to collect data for various patient groups based on age, sex, and disease status, as this may help to determine whether our method can be used to improve diagnosis.” (page8, line257 - page9, line266)

”As such, we were unable to investigate the effects of aging and disease on tongue characteristics. In addition, muscle fibrosis and adiposity are major characteristics of sarcopenia, and in general, age-related fibrosis of muscle tissue is also associated with increased EI [42]. In our method, we categorized the participants’ tongue using EI and TT. Therefore, the influence of aging on EI could not be eliminated. Longitudinal studies that include participants with poor tongue function, systemic sarcopenia, and various diseases may aid in determining the usefulness of our classification method.” (page10, line297-303)

Comment 3.

The diagnostic difficulty with ultrasonography is that the evaluation of Echo intensity is subjective to the evaluator. Please discuss the difficulties in distinguishing between Echo intensity changes due to muscle disease and disuse conditions where active training is recommended.

Response to comment:

Thank you for your comment. Since the values of EI vary depending on the method of measurement and the devices, no index has been established so far to distinguish between healthy and unhealthy states. Most of the studies are based on intergroup comparisons, and as you pointed out, it is a subjective evaluation.

As for the diagnosis of diseases, it is indicated that the EI of the tongue is significantly higher in patients with amyotrophic lateral sclerosis(ALS) (reference No. 35) and muscular dystrophy than in healthy participants. However, its application to screening for disease and disuse conditions is difficult at this stage, and we believe that more data is needed to construct an algorithm. We have revised the “Discussion” (page8, line257 - page9, line266) and “4.5 Limitations” (page10, line297-303).

Comment 4.

There is no "4. Discussion". Please structure it properly. Is "6. Conclusion" after "4.5 Limitations" a mistake for "5. Conclusion"?

Response to comment: 

Thank you for your comment. We have revised the manuscript and provided the appropriate numbers and titles. As for English language and style, since it was pointed out that “Extensive editing of English language and style” is required, the manuscript has been proofread by a native speaker of English again and revised.

Reviewer 2 Report

Dear Authors,

congratulations for the study and for the data gathered so far. The  methods are appropriate and well described, and adequate details are provided to replicate the work.  The discussion is well

balanced and adequately supported by the data. The paper is clearly written and I believe it should be accepted after some minor revisions. Indeed the paper is very interesting, but I have some comments to suggest for the manuscript improvement:

Discussion: 

  • Clinical implications: in lines 265-266 it isn’t the decrease in muscle with age that causes dysphagia, however, the nervous component must be taken in consideration, such as the fibres of the XII cranial nerves.
  • To be more complete, since the age also influence the thickness of the papillae (https://pubmed.ncbi.nlm.nih.gov/21711389/), a mention of the biofilm coating the dorsum of the tongue and the related condition caused by the microbial population is worthy (https://pubmed.ncbi.nlm.nih.gov/31034083/; https://oajournals.fupress.net/index.php/ijae/article/view/3424/3421)
  • Conclusion:

       The conclusions could be argued in more detail.

Author Response

Comments for Reviewer 2

Dear Authors,

congratulations for the study and for the data gathered so far. The  methods are appropriate and well described, and adequate details are provided to replicate the work.  The discussion is well balanced and adequately supported by the data. The paper is clearly written and I believe it should be accepted after some minor revisions. Indeed the paper is very interesting, but I have some comments to suggest for the manuscript improvement:

Comment 1.

Discussion: Clinical implications: in lines 265-266 it isn’t the decrease in muscle with age that causes dysphagia, however, the nervous component must be taken in consideration, such as the fibres of the XII cranial nerves.

Response to comment: 

Thank you for your comment. The aim of this study was to detect oral hypofunction in healthy older adults using an algorithm. Therefore, we only mentioned about the effect of aging in the  “Clinical implications” section. However, as you pointed out, aging is not the only factor that causes dysphagia. Reviewer 1 also pointed out that there are other effects besides aging, and since the expression in the manuscript is not conducive to misunderstanding, the “Discussion” and “4.4 Clinical implications” have been revised as follows.

”Several studies have investigated whether EI can be used for tongue classification. The EI of the tongue increases with age [34]. Research has also indicated that the EI of the tongue is significantly higher in patients with amyotrophic lateral sclerosis (ALS) than in healthy participants [35], suggesting that EI can be used to distinguish between healthy and diseased tongue muscle. However, no previous studies have examined these associations using detailed classifications based on ultrasonography or significant differences in tongue function to define each group. Our findings indicate that K-means clustering may be an effective approach for classification of the tongue. Future studies should aim to collect data for various patient groups based on age, sex, and disease status, as this may help to determine whether our method can be used to improve diagnosis.” (page8, line257 - page9, line266)

”Thus, assessments of muscle mass and the strength/function of the perioral muscles are necessary for maintaining oral function in older adults.” (page9, line269-271)

”Establishing standard diagnostic criteria based on ultrasonography patterns may represent a more useful and non-invasive strategy for assessing and preventing oral hypofunction.” (page9, line272-274)

Comment 2.

To be more complete, since the age also influence the thickness of the papillae (https://pubmed.ncbi.nlm.nih.gov/21711389/), a mention of the biofilm coating the dorsum of the tongue and the related condition caused by the microbial population is worthy (https://pubmed.ncbi.nlm.nih.gov/31034083/; https://oajournals.fupress.net/index.php/ijae/article/view/3424/3421)

Response to comment:

Thank you for your suggestion about the atrophy of tongue papillae with aging and its effect on oral hygiene. We have added the information as follows in the “Clinical implications” section.

“Furthermore, aging leads to atrophy of the tongue papillae, and several studies have reported an association between tongue function and the characteristics of the tongue surface [40,41]. Appropriate diagnostic assessment of tongue function can aid clinicians in selecting the proper treatment strategy, which may in turn aid older adults in maintaining oral hygiene, feeding ability, and swallowing function.” (page9, line282-286)

Comment 3.

Conclusion: The conclusions could be argued in more detail.

Response to comment:

Thank you for your comment. We have summarized the results of this study and describes the Conclusions in detail. The revised part of the manuscript has been highlighted in yellow.

“We used K-mean clustering algorithms……for various ages and diseases in the future.” (page10, line305-318)

Reviewer 3 Report

The research is interesting, however the N is small, in the limitations the authors themselves consider that it might not be enough to give a result without error.

I suggest that the conclusions be extensive, at least half a page.

Author Response

Comments for Reviewer 3

The research is interesting, however the N is small, in the limitations the authors themselves consider that it might not be enough to give a result without error.

I suggest that the conclusions be extensive, at least half a page.

Response to comment:

Thank you for your comment. As you pointed out, more samples are needed to establish the algorithm.

We have revised the “Conclusions” by describing in more detail. The revised part of the manuscript have been highlighted in yellow.

“We used K-mean clustering algorithms……for various ages and diseases in the future.” (page10, line305-318)

Round 2

Reviewer 1 Report

Thanks for the revisions.

No multivariate analysis was performed. It was also written that the effect of age-related changes has not been examined, which is a limitation of this paper. I understood that this paper was intended to show the relationship between TT and EI in a healthy elderly population and to distinguish between groups of EI that suggest a decline in tongue function. In this article, the authors have only presented a classification that is useful for training plans to improve disuse changes in healthy elderly people. In line 279, the authors refer to "(exercises) for specific patients," but I think "for older individuals" or "for community-dwelling older adults" would be more appropriate.

In Reference 39, the authors cite the effects of training on elderly people who need long-term care, but I think this citation is inappropriate because it is written in a way that deviates from the target. Reference 39 should be properly cited, as it appears to be targeted at community-dwelling older people.

In the limitations, please specify that the classification given in this paper cannot detect neuromuscular diseases. The limitation paragraphs need to reorganize the text.

Reviewer 3 Report

I consider it an interesting study, in general it is clear to me. 
The only suggestion is to include the metrics obtained in conclusions and in the summary.
